

# Interferon-$\gamma$ responses to *Plasmodium falciparum* vaccine candidate antigens decrease in the absence of malaria transmission

Cyrus Ayieko[1,*], Bilha S. Ogola[2,*], Lyticia Ochola[3], Gideon A.M. Ngwena[4], George Ayodo[5], James S. Hodges[6], Gregory S. Noland[7] and Chandy C. John[8]

[1] Department of Zoology, Maseno University, Maseno, Kenya
[2] Department of Biological Sciences, Masai Mara University, Narok, Kenya
[3] Department Biological Sciences, Kabianga University, Kericho, Kenya
[4] School of Medicine, Maseno University, Maseno, Kenya
[5] School of Health Sciences, Jaramogi Oginga Odinga University of Science and Technology, Bondo, Kenya
[6] School of Public Health, University of Minnesota, Minneapolis, MN, United States
[7] Medical School, University of Minnesota, Minneapolis, MN, United States
[8] Medical School, Indiana University, Indianapolis, IN, United States
[*] These authors contributed equally to this work.

Corresponding author
Cyrus Ayieko, cxayk@yahoo.com

## ABSTRACT

**Background**. Malaria elimination campaigns are planned or active in many countries. The effects of malaria elimination on immune responses such as antigen-specific IFN-$\gamma$ responses are not well characterized.

**Methods**. IFN-$\gamma$ responses to the *P. falciparum* antigens circumsporozoite protein, liver stage antigen-1, thrombospondin-related adhesive protein, apical membrane antigen-1, MB2, and merozoite surface protein-1 were tested by ELISA in 243 individuals in highland Kenya in April 2008, October 2008, and April 2009, after a one-year period of interrupted malaria transmission from April 2007 to March 2008.

**Results**. While one individual (0.4%) tested positive for *P. falciparum* by PCR in October 2008 and another two (0.9%) tested positive in April 2009, no clinical malaria cases were detected during weekly visits. Levels of IFN-$\gamma$ to all antigens decreased significantly from April 2008 to April 2009 (all $P < 0.001$).

**Discussion**. Naturally acquired IFN-$\gamma$ responses to *P. falciparum* antigens are short-lived in the absence of repeated *P. falciparum* infection. Even short periods of malaria interruption may significantly decrease IFN-$\gamma$ responses to *P. falciparum* antigens.

## INTRODUCTION

More than 200 million people each year develop clinical malaria, mostly in sub-Saharan Africa (*WHO, 2012*). Although scaling up control measures has effectively reduced morbidity and mortality in many African countries (*Feachem et al., 2010b*; *Murray et al., 2012*), emerging cases of drug and insecticide resistance (*Cheeseman et al., 2012*; *Reddy et al., 2013*) necessitate intensified efforts to develop vaccines. The observation that cumulative

exposure to malaria leads to development of partial clinical immunity in malaria endemic areas (*Doolan, Dobano & Baird, 2009*) provides a basis for vaccine development. However, the host-parasite interaction that leads to development of immunity is not well understood. For instance, even after life-long exposure to malaria, many people living in malaria endemic areas have asymptomatic *Plasmodium* infection and poor responses to common *P. falciparum* antigens (*Egan et al., 1996*; *Owusu-Agyei et al., 2001*). Moreover, evidence from mathematical model suggest that interventions that interrupt malaria infection may lead to loss of immunity (*Ghani et al., 2009*). These observations suggest inefficiencies in the process of maintaining protective immunity to malaria antigens.

The role of IFN-$\gamma$ in mediating protection against pre-erythrocytic malaria infection is well demonstrated in murine studies (*Wang et al., 1996*) but less certain in naturally exposed populations (*John et al., 2000*). Some studies have associated IFN-$\gamma$ responses to *P. falciparum* pre-erythrocytic or blood-stage antigens with protection from re-infection (*Luty et al., 1999*), uncomplicated disease (*John et al., 2004*; *Robinson et al., 2009*) and malarial anaemia (*Ong'echa et al., 2003*). Notably, protection induced by the circumsporozoite protein (CSP)-based vaccine, RTS,S is associated with sustained production of IFN-$\gamma$ (*Sun et al., 2003*) and elevated counts of both CSP-specific IFN-$\gamma$-producing as well as multifunctional CD4+ T cells (*Kester et al., 2009*). However, IFN-$\gamma$ responses did not correlate to protection from re-infection in malaria-exposed volunteers after a *P. falciparum* eradication therapy (*Kurtis et al., 1999*) or in malaria-naïve volunteers that participated in Phase II trial of liver stage antigen-1 (*Cummings et al., 2010*). Indeed, the findings of some studies suggest that the breadth and magnitude of IFN-$\gamma$ responses reduce quickly upon resolution of infection (*John et al., 2004*; *Riley et al., 1993*) leading to the suggestion that IFN-$\gamma$ responses may be a marker of recent exposure. In contrast, repeat-cross-sectional studies in highland Kenya showed that IFN-$\gamma$ responses to pre-erythrocytic antigens LSA-1 (*John et al., 2000*) and TRAP were stable over periods of very low malaria transmission (*Moormann et al., 2009*), possibly showing that IFN-$\gamma$ responses may not be a good indicator of past exposure. Interestingly, a recent study suggested that IFN-$\gamma$ responses to blood stage antigens are a marker of recent exposure but not protection, while IFN-$\gamma$ responses to pre-erythrocytic antigen driven are associated with protection (*Jagannathan et al., 2015*). It is therefore not clear how dynamics of antigen-driven IFN-$\gamma$ responses change with antigen in different epidemiological settings and how this may relate to immunity to malaria.

Malaria elimination campaigns aiming to "shrink the malaria map" are currently planned or active in several African and Asian countries (*Feachem et al., 2010a*). However, the numerous resurgence events following incomplete local interruption (*Cohen et al., 2012*) and the risk of re-importation from neighboring endemic areas (*Cotter et al., 2013*) highlight the threat malaria will pose to areas achieving interrupted transmission. The effects of malaria elimination on *P. falciparum* antigen-specific immune responses associated with protection, such as antigen-specific IFN-$\gamma$ responses, have not been studied. Studying such responses is relevant to understanding how changes in immune response may affect risk of susceptibility to malaria re-exposure. As malaria transmission decreases and vaccines are considered as a component of elimination campaigns or to sustain elimination,

assessment of immune responses such as antigen-specific IFN-$\gamma$ responses is also relevant to considerations of vaccine longevity in areas in which malaria transmission is reduced or absent.

We have conducted malaria epidemiology studies for the past decade in an area of Kenya with unstable malaria transmission. After repeated, high-coverage annual spraying of households with indoor residual insecticides, malaria transmission appeared to be interrupted from April 2007 to March 2008 (*John et al., 2009*). Subsequently, a small number of cases have occurred in this area, but the seasonal peaks of transmission observed before 2007 have not reoccurred. In April 2008, we enrolled a cohort of individuals of all ages to assess immune responses to *P. falciparum* in a period of low or absent transmission. During the one-year follow-up period, none of the 234 individuals had clinical malaria as determined by active surveillance although only three (1.3%) had asymptomatic *P. falciparum* infection by polymerase chain reaction (PCR) technique. This cohort provided a unique opportunity to assess IFN-$\gamma$ responses to *P. falciparum* antigens longitudinally in a population that simulated recent interruption of malaria transmission.

## MATERIALS AND METHODS

### Study population and malaria surveillance

The study was conducted in the adjacent highland locations of Kipsamoite and Kapsisiywa in North Nandi District in western Kenya, which have historically had unstable malaria transmission (*Ernst et al., 2006*; *John et al., 2005*). Intensified annual indoor residual spraying of long-lasting insecticides and introduction of artemether-lumefantrine as first-line therapy for symptomatic malaria produced a period of apparent interrupted malaria transmission from April 2007 to March 2008 in which no clinical cases of microscopy-confirmed malaria were recorded on site (*John et al., 2009*).

Written informed consent was obtained from the participants or their guardians (in case of children). The study was approved by the ethical and scientific review committees at the Kenya Medical Research Institute and the Institutional Review Board at the University of Minnesota (SSC Protocol No. 939).

Three hundred and five individuals, randomly selected (from a total population ~8,000 in 2008) over a one-year period, consented to participate in the longitudinal study. Healthy individuals over the age of two years who had lived at the site for more than six months were eligible for inclusion. Samples were collected in April 2008, October 2008, and April 2009. Two hundred and forty-three of the 305 individuals were present at all three time points and had sufficient peripheral blood mononuclear cells (PBMC) for testing of IFN-$\gamma$ responses to all six antigens to be studied (mean age 25.9, median age 22.3, age range 2.4 years to 82 years), though IFN-$\gamma$ responses to individual antigens are missing at some time points.

Clinical malaria was assessed by weekly active surveillance of study participants by trained village health workers for malaria symptoms (fever, headache, chills, severe malaise). Participants with symptoms of malaria were referred to the local dispensary for further evaluation and treatment.

Blood samples obtained in April 2008, October 2008, and April 2009 were also tested for asymptomatic *P. falciparum* parasitemia by microscopy (*John et al., 2005*) and nested PCR (*Menge et al., 2008*) as previously described (*Noland et al., 2012*). Further assessment for malaria transmission was by measuring prevalence of antibodies to *P. falciparum* antigens CSP (*Esposito et al., 1988*) and schizont extract using ELISA previously described (*Wipasa et al., 2010*).

## Blood sample collection and processing

Venous blood (3–5 mls from children and 10–15 ml from adults) was collected into heparinized vacutainers (BD, Plymouth, UK) separated by histopaque density gradient centrifugation. PBMC were cultured in 96 well plates and stimulated with *P. falciparum* peptides. Phytohemagglutinin (PHA) and PBS were used as positive and negative controls respectively.

## Selection of peptides of *P. falciparum* pre-erythrocytic and blood stage antigens

*P. falciparum* pre-erythrocytic and blood stage antigens were selected for testing based on prior studies demonstrating immunogenicity or association with protection from *P. falciparum* infection and disease in malaria endemic areas. All antigens tested except MB2 have been assessed as vaccine candidate antigens. Peptides selected for testing are detailed in Table 1. They included peptides from the pre-erythrocytic antigens: circumsporozoite protein (CSP (*Reece et al., 2004*)), liver stage antigen-1 (LSA-1), thrombospodin-related adhesive protein (TRAP) (*Flanagan et al., 1999*), and the blood-stage antigens: apical membrane antigen-1 (AMA-1) (*Lal et al., 1996*; *Udhayakumar et al., 2001*), MB2 (pool of peptides MB1 (*Nguyen & James, 2001*) and MB2, (*Nguyen & James, 2001*)), and merozoite surface protein-1 (MSP-1, pool of peptides M1 (*Parra et al., 2000*) and M2, (*Udhayakumar et al., 1995*)). The peptides were synthesized and purified by high-performance liquid chromatography (HPLC) to >95% purity (Sigma Genosys, St. Louis, MO, USA).

## Antibody ELISA testing

To detect any evidence of recent malaria exposure in the population, plasma samples were tested for antibodies to circumsporozoite protein (CSP) and schizont extract by ELISA. The $(NANP)_5$ repeat peptide was used for the circumsporozoite protein (CSP) while the schizont extract was obtained from cultures of 3D7 *P. falciparum* strain. Both CSP peptide and schizont extract were were dissolved in 0.01 M PBS to a concentration of 10 µg/ml. Briefly, 50 µl of antigen solution was added to Immulon-4HBX plates (Dynex Technologies, Chantilly, VA, USA). Following overnight incubation at 4 °C, and blocking in 5% nonfat powdered milk in PBS, 50 µl samples of serum diluted at 1:100 in 5% powdered milk were added to wells and incubated for 2 h at room temperature. After washing with PBS −0.05% Tween 20, 50 µl of alkaline phosphatase conjugated goat anti-human IgG (Jackson ImmunoResearch, West Grove, PA, USA) diluted 1:1,000 in 5% powdered milk was added and incubated for 1 h. After extensive washing with PBS-0.05% Tween 20, p-nitrophenylphosphate was added in accordance with the manufacturer's instructions (S0942; Sigma, St. Louis, MO, USA). The optical density (OD) was measured at 405 nm
**Table 1  Peptides used for IFN-γ testing.**

| | Peptide | Amino acid (sequence) | Reference |
|---|---|---|---|
| Pre-erythrocytic antigens | Circumsporozoite protein (CSP), peptide cs22 | amino acid 378 to 392 (DIEKKICK-MEKCSSV) | *Reece et al. (2004)* |
| | Liver Stage Antigen-1 (LSA-1), peptide T3 | Amino acid 1813 to 1835 (HNENLD-DLDGIEKSSEELSEEKIOH) | *Flanagan et al. (1999)* |
| | Thrombospodin-Related Adhesive Protein (TRAP), peptide tp6 | Amino acid 51 to 70 (HLLMDCSGSIR-RHNWVNHAVPOH) | *Flanagan et al., (1999)* |
| Blood stage antigens | Apical Membrane Antigen-1 (AMA-1), peptide PL171 | aa 348 to 366, aa sequence DQPKQYE-QHLTDYEKIKEG | *Lal et al. (1996)*, *Udhayakumar et al. (2001)* |
| | MB2, a pool of two peptides: | | |
| | MB1 | Amino acid 191 to 199, (SVSSINTNL) | *Nguyen & James (2001)* |
| | MB2 | Amino acid 119 to 127 (KPKKKYYEV) | *Nguyen & James (2001)* |
| | Merozoite Surface Protein-1 (MSP-1), pool of two peptides: | | |
| | M1 | Amino acid 20 to 39, (VTHESYQELVKKLEALEDAV) | *Parra et al. (2000)* |
| | M2 | Amino acid 1467 to 1483, (GISYYEKVLAKYKDDLE) | *Udhayakumar et al. (1995)* |

(Molecular Devices, Sunnyvale, CA, USA). Antibody levels were expressed in arbitrary units (AUs), calculated by dividing sample OD for test plasma sample by the mean plus 3 SD from plasma from malaria naïve North American controls. An AU >1.0 was considered a positive response.

## Cell culture and ELISA for cytokine testing

PBMC was diluted in RPMI media (containing 10% human AB serum, 10 mM L-glutamine, 10 μg/mL of gentamicin) at a concentration of $10^6$ cell/ml then plated in duplicate at $2 \times 10^5$ PBMC/well in round-bottomed plates (Corning® Costar® microtiter plates). *P. falciparum* antigens were added at a concentration of 10 μg/ml while the PHA positive control was added at 5 μg/ml. Cells were incubated at 37 °C, 5% $CO_2$ and humidity for ∼120 h. After which, supernatants were harvested and stored −20 °C for later batch analysis. IFN-γ responses were measured by ELISA assay as previously described (*John et al., 2004*). To obtain the cut-off, PBMCs from 13 malaria-naïve North American controls were cultured in similar conditions as test samples above. The culture supernatants were collected after 120 h incubation and tested for IFN-γ. A positive IFN-γ response to an antigen (cut-off) was defined as a response >mean + 2 standard deviations of the IFN-γ responses of 13 North American individuals never exposed to malaria. Cut-off IFN-γ levels (pg/ml) for individual antigens were: CSP, 43; LSA-1, 101; TRAP, 27; AMA-1, 49; MB2, 67; MSP-1, 31.

## Statistical analysis

Prevalence of IFN-γ responses was compared between age groups using Pearson's $\chi^2$ test; change between two time points was tested using the exact two-tailed McNemar test.

In analyzing levels of IFN-γ responses, an antigen's dependent variable was the common (base 10) logarithm of its IFN-γ response. To include zero responses, a number was added

to the levels before taking the log; for each antigen, this number was the 2.5th percentile of positive IFN-$\gamma$ levels. To test change over time in log IFN-$\gamma$ response for an antigen, a mixed linear model was used. Site and site-by-time interaction were included to allow differences between sites, but these were not significant and are not discussed further. Preliminary analyses also found that age and sex had negligible effect on the results, so they were excluded from the analyses reported here. Pair wise comparisons of times used Tukey's post-hoc test separately for each antigen. The mixed linear model analysis was checked using a parametric bootstrap (sampling persons), which gave nearly identical $P$-values.

The half-life of IFN-$\gamma$ response was estimated using the method of *Wipasa et al. (2011)*. The analysis used a mixed linear model with a random intercept for each individual and fixed effects for site (Kipsamoite or Kapsisiywa) and for time, treated as a continuous measure. Half-life was estimated as follows. The mixed linear model used to analyze the data was

$$\log(\text{IFN} - \gamma \text{ level at time } t) = \text{intercept} + \alpha \times t + [\text{other items}] + \text{error},$$

where time $t$ measures time from time 0—for each person, time 0 was the collection date in the April 2008 wave—with units "years" and the "other items" are constant for a given person. The coefficient $\alpha$ estimates the decline per year in log(IFN-$\gamma$ level). The half-life is the time t such that $0.5 = $ IFN-$\gamma$ level at time $t$/IFN-$\gamma$ level at time 0; taking the common logarithm of both sides of the latter equation gives

$$-\log(2) = \log(\text{IFN} - \gamma \text{ level at time } t) - \log(\text{IFN} - \gamma \text{ level at time } 0) = \alpha \times t.$$

Solving for $t$ gives the half-life as $-\log(2)/\alpha$, and half-life was estimated by replacing $\alpha$ with its estimate. The 95% confidence interval for half life was computed by transforming the upper and lower limits of $\alpha$'s confidence interval; bootstrap confidence intervals were nearly identical.

Analyses used Stata 12 (Stata Corp, College Station, TX, USA) and the R system (*R Development Core Team, 2012*), specifically the lme function in the lme4 package (*Bates, Maechler & Bolker, 2011*).

## RESULTS

### Study population characteristics and incidence of *P. falciparum* infection and disease

The age distribution of the 243 study participants was <5 years, $n = 22$ (9.1%); 5 to 14 years, $n = 73$ (30.0%); 15 to 40 years, $n = 94$ (38.7%); and >40 years, $n = 54$ (22.2%). There were 118 females (48.5%) and 125 males (51.4%). None tested positive for *P. falciparum* infection by microscopy during the sample collections in April 2008, October 2008 or April 2009 (when all were asymptomatic). While none was PCR positive for *P. falciparum* in April 2008, one individual (0.4%) tested positive in October 2008 and another two (0.9%) in April 2009. All were part of the area-wide passive surveillance for clinical malaria from March 2007 to April 2008, and none developed clinical malaria during

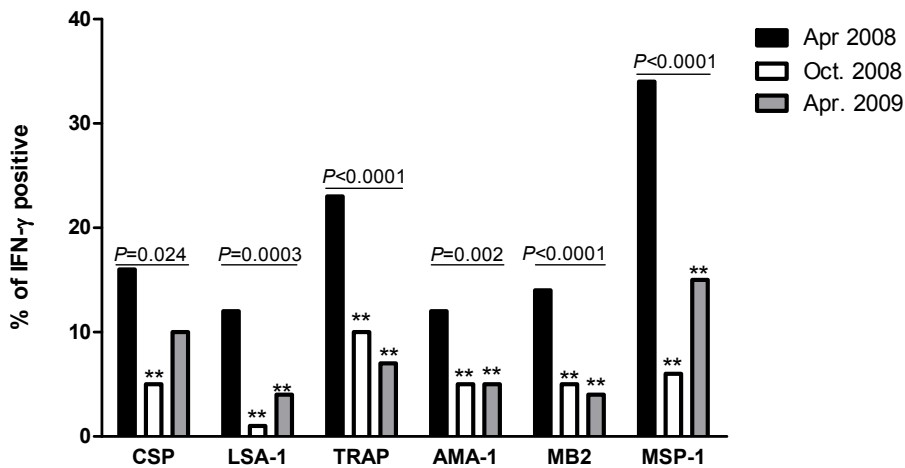

**Figure 1** **Proportion of individuals with positive IFN-$\gamma$ responses to *P. falciparum* antigens in April 2008, October 2008 and April 2009.** The prevalence of antigen-specific responses across time was assessed by the chi-square test for trend. The *P* values are indicated across the bars. Bars marked with double asterisks (**) differ significantly ($P < 0.05$) from baseline (April 2008).

the one-year period of study as determined by weekly active surveillance by village health workers.

Intensity of malaria transmission was, further, assessed by serology (*Drakeley et al., 2005*; *Esposito et al., 1988*). Between April 2008 and April 2009, prevalence of seropositivity of IgG antibodies to CSP remained unchanged (April 2008, 8.3%; April 2009, 4.5%; $P = 0.11$) while prevalence of seropositivity of antibodies to schizont extract decreased significantly (April 2008, 7%; April 2009, 0%; $P < 0.0001$). Antibody levels to both CSP and schizont extract decreased significantly over the period (Table S1). The implication was that no recent malaria transmission had occurred at the two sites over the study period.

## Prevalence of IFN-$\gamma$ responders to *P. falciparum* antigens decreases in the absence of malaria transmission

The prevalence of IFN-$\gamma$ responders to *P. falciparum* antigens decreased significantly for all six antigens between April 2008 and October 2008 (Fig. 1). Between October 2008 and April 2009, prevalence further declined or remained the same for TRAP, AMA-1 and MB2, while non-significant increases occurred for CSP, LSA-1 and MSP-1. Compared to April 2008, prevalence of IFN-$\gamma$ responders in April 2009 remained significantly lower for all antigens except CSP, which was of border-line significance ($P = 0.06$).

## IFN-$\gamma$ levels to *P. falciparum* antigens decrease significantly in the absence of transmission

IFN-$\gamma$ levels to all *P. falciparum* antigens decreased significantly between April 2008 and April 2009 (all $P \leq 0.001$, Fig. 2). However the pattern of decrease depended on antigen, with IFN-$\gamma$ levels to CSP, MB2 and MSP-1 decreasing significantly between April and October 2008 but not between October 2008 and April 2009, IFN-$\gamma$ levels to LSA-1 similar between April and October 2008, then decreasing significantly between October 2008 and April 2009, and IFN-$\gamma$ levels to AMA-1 and TRAP decreasing at both time points (so the

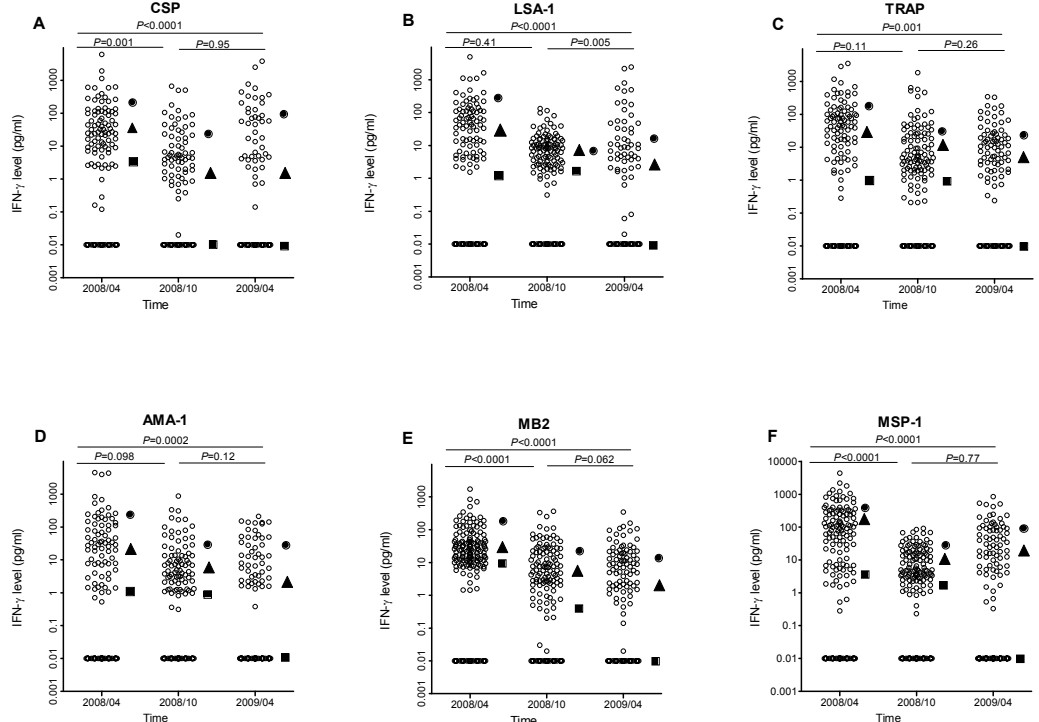

**Figure 2** **IFN-γ levels in response to *P. falciparum* antigens at baseline (April 2008), October 2008, and April 2009.** For each antigen, only individuals with a positive response in at least one time point were included in the analysis. The respective *N* values for the antigens were 144, 165, 169, 142, 194 and 183, respectively, for CSP, LSA, TRAP, AMA, MB-2 and MSP-1. The median IFN-γ level at each time is indicated by a square, 90th percentile by a circle, and 75th percentile by a triangle. Non-responders are placed below responders by giving them an artificial level of 0.01 and then ''jittering'' their levels (adding random noise) vertically to give an accurate sense of the number of non-responders.

**Table 2** **Estimated half-lives of the levels of IFN-γ responses to *P. falciparum* antigens.**

| Antigen | N used to calculate IFN-γ half-life | IFN-γ half life (days) | |
|---|---|---|---|
| | | Estimate | 95% confidence interval |
| CSP | 243 | 157.0 | 113.2 to 255.5 |
| LSA-1 | 242 | 197.1 | 135.1 to 343.1 |
| TRAP | 242 | 259.2 | 171.6 to 525.6 |
| AMA-1 | 242 | 230.0 | 157.0 to 427.1 |
| MB2 | 229 | 138.7 | 113.2 to 175.2 |
| MSP-1 | 239 | 153.3 | 116.8 to 226.3 |

overall decrease from April 2008 to April 2009 was statistically significant, though neither individual decrease was). Using the data from the three time points, the estimated half-lives in years for the level of IFN-γ responses for the different antigens ranged from 0.42 to 0.71 years (Table 2).

Given that sample testing for each time point was done separately from the other time points, the study also investigated whether differences in experimental condition could

have led to the observed differences IFN-$\gamma$ levels. Thirty randomly selected samples were tested such that all the three time-points were in the same plate. A decrease in IFN-$\gamma$ responses was observed for CSP, LSA-1, TRAP, MB2 and MSP-1 between April 2008 and October 2008 but between October 2008 and April 2009 the levels remained similar. In contrast, levels of IFN-$\gamma$ responses to AMA-1 did not vary between the time points (Table S2). The decrease in levels of IFN-$\gamma$ to CSP, LSA-1, TRAP, MB2 and MSP-1 over time could, therefore, be attributed to decrease in malaria transmission rather than batch effect. However, the trend of IFN-$\gamma$ responses to AMA-1 may partly have been due to batch effect.

### IFN-$\gamma$ prevalence and levels to *P. falciparum* antigens differ by antigen but not by age

The proportion of IFN-$\gamma$ responders to *P. falciparum* antigens in April 2008, the period of highest IFN-$\gamma$ responses, was similar for CSP (15.6%), LSA-1 (11.5%), AMA-1 (12.4%) and MB-2 (14.0%), but lower than the proportions of responses to TRAP (23.1%) and MSP-1 (33.8%) ($P = 0.0028$ for TRAP and $P < 0.001$ for MSP-1 compared to AMA-1). Frequency of IFN-$\gamma$ responses to *P. falciparum* antigens did not differ by age, except for responses to MSP-1, which similar in frequency across ages in October 2008, but higher in frequency in younger individuals in April 2009 and in older individuals in October 2009 (Fig. 3). IFN-$\gamma$ levels were not significantly associated with age at any measurement time, except LSA-1 in October 2008 (Pearson's $r = 0.19$; $P = 0.0034$) and MSP-1 in April 2009 (Pearson's $r = 0.16$; $P = 0.0131$) (Fig. S1).

## DISCUSSION

The present study demonstrates that IFN-$\gamma$ responses to several pre-erythrocytic and blood stage *P. falciparum* candidate vaccine antigens decrease in the absence of malaria transmission. The study area and cohort provided a unique opportunity to study IFN-$\gamma$ responses longitudinally in a situation like malaria elimination: interruption of malaria transmission in the study area occurred in the year before the study. No one in the study cohort had an episode of clinical malaria in either the prior year or the study year and the prevalence of asymptomatic parasitemia (measured by PCR) was 0%, 0.4% and 0.9%, respectively, in April 2008, October 2008 and April 2009. Assessment of malaria transmission by serology also showed that IgG levels to CSP and schizont extract decreased significantly over the study period. The implication was that no recent malaria transmission had occurred at the site over the study period. IFN-$\gamma$ responses to several of the antigens studied, including CSP (*Chuang et al., 2013*), LSA-1 (*John et al., 2004*), TRAP (*John et al., 2004*) and MSP-1 (*Moormann et al., 2013*), are associated with protection from *P. falciparum* infection and disease. Moreover, IFN-$\gamma$ responses to CSP (*Sun et al., 2003*) and TRAP (*Ogwang et al., 2013*) were associated with protection from disease in studies of vaccines containing these antigens. In populations like those in this highland area, which is surrounded by areas of high malaria transmission, loss of these IFN-$\gamma$ responses may increase susceptibility to clinical disease should infection be reintroduced to the area (e.g., as a consequence of increased travel, climate change, or drug or insecticide resistance).

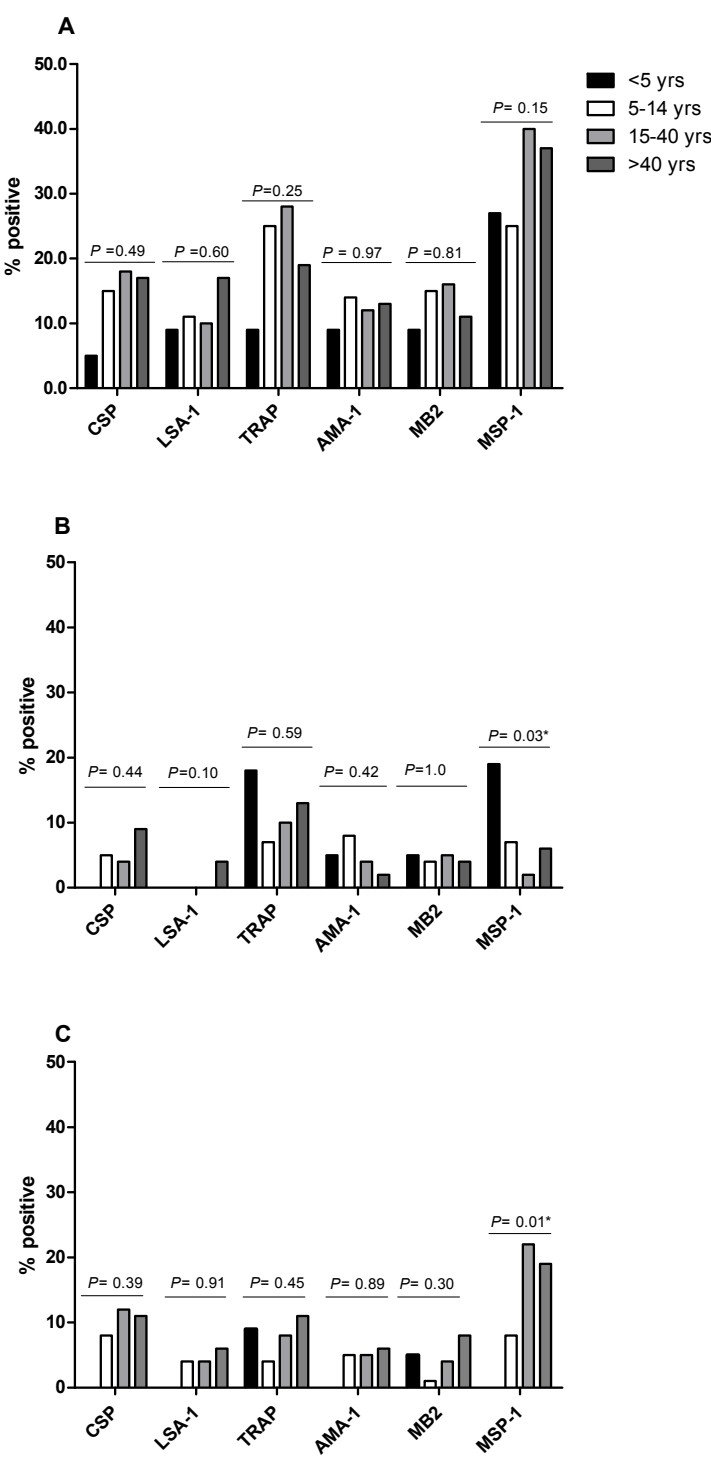

**Figure 3** **Proportion of individuals with positive IFN-$\gamma$ responses to the *P. falciparum* antigens CSP, LSA-1, MB2, TRAP, AMA-1, MB2 and MSP-1 across ages in April 2008 (A), October 2008 (B) and April 2009 (C).** IFN-$\gamma$ responses across age-groups were compared by Fischer's exact test. The *P* values for each antigen are indicated.

The present results are also relevant to campaigns in which vaccines are part of a malaria elimination strategy. Although malaria vaccines may induce strong and more long-lived responses to *P. falciparum* antigens (*Teirlinck et al., 2011*) than natural infection (*Dent et al., 2009*; *Migot et al., 1993*; *Wipasa et al., 2011*), these results suggest that vaccine-boosting considerations should be an important part of planning such campaigns.

Previous studies documented decreases in IFN-$\gamma$ responses to *P. falciparum* schizont extract in Thai adults in an area of very low transmission (*Wipasa et al., 2011*) and to ring-infected erythrocyte surface antigen (RESA) in adults and children in Madagascar several years after a malaria epidemic (*Migot et al., 1993*). However, both schizont extract and RESA are not established targets of protective immunity. In the present study, we documented that IFN-$\gamma$ responses decrease for all ages and all *P. falciparum* antigens tested. The lack of age-related increase in IFN-$\gamma$ responses in this study contrast with previous findings (*Chelimo et al., 2011*; *John et al., 2004*; *Rolfes et al., 2012*). It is notable that Chelimo et al. was conducted their study in a malaria holoendemic area where malaria transmission was intense. The studies by John et al. and Rolfes et al. were both conducted in highland Kenya, before interruption of malaria transmission in 2008 (*John et al., 2009*). The finds suggest that in the absence of frequent and repeated exposure to malaria antigens IFN-$\gamma$ responses reduce to baseline levels in both children and adults, hence loss of age-wise relationship. IFN-$\gamma$ levels to most antigens decreased most dramatically in the first six months of the study and then remained at the lower level in the latter six months, suggesting a biphasic pattern of reduction of IFN-$\gamma$ responses to malaria antigens. Similar findings have been recorded for anti-circumsporozoite antibodies after vaccination (*White et al., 2015*). However, a few individuals developed or maintained high levels (>100 pg/ml) in April 2009, suggesting that immune responses remain vigorous in a subset of individuals or that occasional variations can occur in measuring these responses. In a prior smaller study we conducted using multiplex cytometric bead assay (CBA) testing for multiple cytokines and chemokines, IFN-$\gamma$ responses to multiple antigens in this population also showed a decrease in IFN-$\gamma$ levels to most antigens at six months, and a trend toward lower levels at 12 months (*Ochola et al., 2014*). That small cohort inspired the testing in the larger cohort of this study, which showed conclusively that IFN-$\gamma$ responses to pre-erythrocytic and blood stage antigens were significantly lower 12 months after possible interruption of transmission. Previous studies we have done in this population and in a population in a malaria holoendemic area also found that measured IFN-$\gamma$ responses were often not consistent over time in a given individual (*Moormann et al., 2006*). These findings suggest limitations and potential day-to-day variability in detection of IFN-$\gamma$ responses to *P. falciparum* antigens in peripheral blood. However, a strength of our study was the large cohort size for this type of testing, which allowed us to detect decreases in IFN-$\gamma$ levels to all antigens, confirming that the decreases overall were real and not due to chance variation. Measurements at three time points also allowed demonstration that for most antigens, values decreased and stayed lower or decreased and continued to decrease further over time. These findings support the hypothesis that IFN-$\gamma$ responses to *P. falciparum* antigens may be a marker of recent malaria exposure.

The estimated half-life of IFN-$\gamma$ responses varied modestly by antigen (0.42 to 0.71 years), but all half-life estimates were under one year and the 95% confidence interval did not exceed 1.5 years for any antigen. The half-lives of IFN-$\gamma$ responses, in the Thailand population, were somewhat longer (estimate of 3.27 years, 95% CI 1.94 to 10.26 years) (*Wipasa et al., 2011*), probably because the authors tested responses to schizont extract, which is a mixture of different antigens, as opposed to single peptides used in this study. It is notable that longer IFN-$\gamma$ half-lives *P. falciparum* sporozoites, infected RBCs (*Roestenberg et al., 2011*; *Todryk et al., 2008*) and blood-stage parasites (*Teirlinck et al., 2011*) after experimental exposure of volunteers to infective mosquito bites were assessed after 24-hour incubation (*Roestenberg et al., 2011*). In the present study, IFN-$\gamma$ responses were assessed from PBMC cultured for 120 h suggesting that the shorter half-lives may be due in part to a difference in what was assessed (*Flanagan et al., 2001*; *Todryk et al., 2009*).

Aggressive implementation of vector control measures, such as residual insecticide spraying and insecticide treated bed nets, has significantly reduced malaria transmission in several countries (*WHO, 2012*). As in highland Kenya, previously endemic populations are likely to experience prolonged periods of low transmission. The present study's results suggest that under such situations, vaccinated individuals are likely to lose vaccine-induced immune responses. Phase II trials of the CSP-containing vaccine RTS,S indicated that protection lasted less than two years in a significant proportion of African children (*Alonso et al., 2005*; *Guinovart et al., 2009*). It is unclear whether efficacy will be briefer or longer in areas of very low transmission, but this study's data suggest it may be briefer, and that malaria elimination campaigns that include vaccination as part of the elimination strategy will need to assess carefully the longevity of putative protective immune responses.

A limitation of this study was that malaria transmission had been interrupted at the site approximately one year prior to the study, hence decline in IFN-$\gamma$ levels may have begun prior to the study. The calculated half-lives of IFN-$\gamma$ responses for each antigen are, therefore, likely to vary from the true antigen-specific IFN-y half-life responses. This warrants future studies involving participants with *P. falciparum* infection at baseline.

## CONCLUSION

Residents of highland Kenya had a significant decrease in IFN-$\gamma$ responses to multiple *P. falciparum* vaccine candidate antigens during a one-year period in which there was no evidence of malaria infection within the cohort. These results suggest that even short periods of transmission interruption can lead to a loss of immune responses. Further studies are warranted to assess the cellular basis for loss of malaria-specific IFN-$\gamma$ as well as the stability of specific responses to other pro- and anti-inflammatory cytokines. Additional studies should also examine the kinetics of immune response re-acquisition following re-exposure in previously immune individuals.

## ACKNOWLEDGEMENTS

We thank the study participants and the field assistants at Kipsamoite and Kapsisiywa villages in Nandi County. We are also indebted to Jackson Abuya, David Koech, the late

Livingstone Wanyama, Bartholomew Ondigo, Kelvin Onoka, Jackline Omundi, Lilian
Kisia and Jonathan Bett for technical assistance in field collections and with the assays.

### Funding
This work was supported by grants from the National Institute of Allergy and Infectious
Diseases (Grant 5U01 A1056270 to CCJ) and the Fogarty International Center (Grant D43
TW008085 to CCJ). The funders had no role in study design, data collection and analysis,
decision to publish, or preparation of the manuscript.

### Grant Disclosures
The following grant information was disclosed by the authors:
National Institute of Allergy and Infectious Diseases: 5U01 A1056270.
Fogarty International Center: D43 TW008085.

### Competing Interests
The authors declare there are no competing interest

### Author Contributions
- Cyrus Ayieko performed the experiments, analyzed the data, wrote the paper, prepared figures and/or tables.
- Bilha S. Ogola conceived and designed the experiments, performed the experiments, analyzed the data, wrote the paper, prepared figures and/or tables.
- Lyticia Ochola performed the experiments, analyzed the data, prepared figures and/or tables, reviewed drafts of the paper.
- Gideon A.M. Ngwena conceived and designed the experiments, performed the experiments, reviewed drafts of the paper.
- George Ayodo reviewed drafts of the paper.
- James S. Hodges analyzed the data, prepared figures and/or tables, reviewed drafts of the paper.
- Gregory S. Noland performed the experiments, wrote the paper, reviewed drafts of the paper.
- Chandy C. John conceived and designed the experiments, contributed reagents/materials/analysis tools, reviewed drafts of the paper.

### Human Ethics
The following information was supplied relating to ethical approvals (i.e., approving body and any reference numbers):

The study was approved by the ethical and scientific review committees at the Kenya Medical Research Institute and the Institutional Review Board at the University of Minnesota (SSC Protocol No. 939).

## Data Availability

The raw data has been supplied as a Supplementary File.

## Supplemental Information

Supplemental information for this article can be found online at http://dx.doi.org/10.7717/peerj.2855#supplemental-information.

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
