# Peer review of "Interferon-γ responses to Plasmodium falciparum vaccine candidate antigens decrease in the absence of malaria transmission"

_PeerJ, doi:10.7717/peerj.2855_

## Round 0.1 · original submission · Minor Revisions

Your manuscript has been assessed by three expert reviewers. Based on their reports, and my own assessment, I am pleased to inform you that it is potentially acceptable for publication, once you have carried out some essential revisions suggested by them. Please provide more details about antibody responses to P. falciparum antigens across the manuscript.

·

Basic reporting

This manuscript address an important but often not reported issue in that it investigates the T cell memory to malaria antigens during a time period of maleria reduction. The basic reporting is sound and the paper is generally well written with all the relevant data included in the manuscript or as supplemental data.

Experimental design

The experimental design is very straightforward and satisfactory as IFN-g expression in response to specific peptide stimulation is a very common approach used towards understanding if a host was exposed to a specific pathogen.

Validity of the findings

The findings are very relevant to the possible strategy for utilizing malaria vaccines as they become available for testing. Conclusions are reasonable, given the data and appropriate speculation about the possible need for boosters is included.

Additional comments

This is a very straightforward paper that investigates the memory of a response to malaria antigens during a period of low malaria transmission. The data is straightforward and consistent with a model that would possibly require booster shots that would strengthen the host immune response. I only have a few comments:

1. The authors state the lack of any sex bias but did they compare the sexes within the different age groups?
2. The population was healthy at the start but were there any later infections that might have impacted the results?
3. How were cutoff IFN levels for individual peptides determined?

Reviewer 2 ·

Basic reporting

There are a number of typographical errors in the manuscript – it is suggested that the authors carefully re-read their manuscript and fix these errors to improve the language and readability. The introduction is otherwise well written, clear and easy to understand. Past literature has been adequately cited, although it is recommended that the more recent version of the WHO world malaria report be used. The structure of the manuscript confirms to PeerJ standards. The figures and tables are relevant; some suggestions for improvement have been made (see General Comments). The raw data has been provided, however further information should be included to make this easier to understand i.e. include another sheet in the excel files that provides a description for what each of the variables/headers means.

Experimental design

Overall, the study has a clear research question on the longevity of antigen-specific IFN-y responses following interrupted malaria transmission. This topic will be of interest to the malaria immuno-epidemiology and vaccine fields.

It would have been preferable if the authors had tested IFN-y levels for each individual at all time-points on the one plate. They have tried to account for this by measuring the batch effect. However, they have failed to discuss the results of this experiment. Given its importance this should be discussed, even if just briefly in the results section.

Half-lives have been calculated from a start-point when there had already been no malaria exposure for approximately one-year. Hence the IFN-y responses may have already declined significantly since the last exposure. This may result in some variability of the estimate compared to the true IFN-y half-life responses for these antigens. I think the methodology for obtaining the estimated half-lives should be introduced in more detail, either in the materials and methods or results sections.

Validity of the findings

The conclusion of the manuscript is well supported by the authors’ findings, and the findings themselves are valid.

Additional comments

Introduction
Line 67-69. If studies have suggested the breadth and magnitude of IFN-y responses reduce quickly upon resolution of infection, it would suggest to me that IFN-y responses are more markers of current or recent exposure, not past exposure (as stated in Line 69). ‘Past exposure’ does not define very well how far in the past you mean.

Materials and methods
Line 113. Should include the ethics approval numbers.
Line 121. Define PBMC, peripheral blood mononuclear cells.
Line 122. Also state the median/mean age.
Line 125. Was clinical malaria assessed weekly during the entire study period, i.e. April 2008-April 2009? This is not clear. Were the individuals with symptoms (referred to the local health facilities) tested for malaria, or only at the three visits?
Line 138. Define PHA. I don’t think the details on PBMC culturing are needed here, given this is explained in detail in section 2.5.
Line 145. Define MB2. Which antigen is this?
Section 2.3. Would the peptide information be simpler to read in a table format?

Results
Line 225. I assume this refers to ‘prevalence of seropositivity’ of IgG antibodies to CSP remained unchanged. This should be clarified.
Line 228. I could not locate supplementary table 1 in the materials provided.
Figure 2. Suggest adding in the positivity cut-offs as a line on each graph.
Table 1. The estimated half-lives could be given in days or months, given they are all less than 1 year.

Discussion
Line 276. The results of the antibody ELISA measurements should be mentioned here, given the reason for doing this was to support the assumption that there was no/limited exposure to malaria parasites during the study period.
Line 305. It is interesting that the responses decreased most dramatically in the first 6 months of the study. This potentially suggests that the decay in IFN-y responses is not simply linear, but maybe biphasic, as has been shown for antibody responses (i.e. White MT et al 2014 BMC Med).
Line 310. Were any of the same samples used? It would be interesting to compare the IFN-y responses tested using the two different assays in the same population.
Line 330. Review this sentence: ‘It is notable…’. The grammar needs to be checked to make sure the meaning is clear.
Line 335. Could this also be an effect of comparing individuals from non-malaria endemic areas and malaria endemic areas? For example, often vaccines perform better in trials of individuals from non-malaria endemic regions.

Reviewer 3 ·

Basic reporting

See comments below.

Experimental design

See comments below.

Validity of the findings

See comments below.

Additional comments

Ayieko et al. aimed to test the specific IFN-gama response in a big cohort of individuals living in an area where malaria transmission was interrupted for more than a year. Three blood collections were performed in one year-interval, and the IFN-g response against six different Plasmodium antigens derived from pre-erythrocytic and erythrocytic phases was tested. During the study period, the incidence of malaria in the area was extremely low (less than 1%). The authors show a significant drop in the % of specific IFN-g positive responses over time.

The paper is well written and is easy to follow. The results add new information on specific IFN-g immune responses in areas where Plasmodium falciparum malaria transmission has been interrupted. This is relevant because IFN-g production has been correlated with protection against malaria, and a drop on this response may lead to increased risk of malaria acquisition, especially in the context of an anti-malaria vaccine. It is important to mention that the authors carefully discuss some articles that reached different conclusions and try to point out the differences among the studies. Overall, this paper adds interesting information on the dynamics of IFN-g responses in areas that may have malaria transmission interrupted for some time.

Minor comments:

1. I could not find Supplementary Table 1 in the submitted material.
2. ELISAs were performed with CSP and schizont extract. Please provide information on both. Which CSP was used? Recombinant protein produced in bacteria? From which Plasmodium clone? What was the concentration added per well? And how was the schizont extract prepared? What was the concentration used in the ELISAs? This information should be added to the materials and methods section.

---

## Round 0.2 · accepted · Accept

The authors have properly addressed all the comments made by the reviewers.

·

Basic reporting

The modifications meet my concerns.

Experimental design

The modifications meet my concerns

Validity of the findings

no comments

Additional comments

no new comments

Reviewer 2 ·

Basic reporting

The addition of the sheet named ‘Log’ to the raw data file is greatly appreciated. I still feel that a final check of the grammar is required, for instance the sentence beginning on Line 393, although this is not required to pass the editorial criteria.

Experimental design

The authors have provided further details of the mathematical model for obtaining half-lives, which will be appreciated by readers. They have also added in the implications of the potential batch effect.

Validity of the findings

The conclusion of the manuscript is well supported by the authors’ findings, and the findings themselves are valid.

Additional comments

The authors have adequately revised their manuscript. Note the authors said they had added a positivity cut-off line for the panels in Figure 2; this isn’t reflected in the Figure 2 I have been sent.

Reviewer 3 ·

Basic reporting

The authors have addressed all the comments made by this reviewer.

Experimental design

No comments.

Validity of the findings

No comments.

Additional comments

No comments.

---

## Author Rebuttal · Round 0.2

# MASENO UNIVERSITY
# SCHOOL OF PHYSICAL AND BIOLOGICAL SCIENCES
# DEPARTMENT OF ZOOLOGY

Private Bag

**MASENO,** KENYA.

**14th November 2016**

Dear Editors,

We would like to thank you for taking time to review our manuscript and for the useful comments and suggestions. We have edited the manuscript accordingly to address your concerns and our specific responses are outlined below.

We hope that you will now find the manuscript suitable for publication.

Thank you.

**Cyrus Ayieko, PhD**

**On behalf of all the authors.**

Editor's Comments

*Your manuscript has been assessed by three expert reviewers. Based on their reports, and my own assessment, I am pleased to inform you that it is potentially acceptable for publication, once you have carried out some essential revisions suggested by them. Please provide more details about antibody responses to P. falciparum antigens across the manuscript.*

**We have added more details of antibody responses on line 164-168, line 244-249,**

*Reviewer 1 (Howard Young)*

## Basic reporting

*This manuscript address an important but often not reported issue in that it investigates the T cell memory to malaria antigens during a time period of maleria reduction. The basic reporting is sound and the paper is generally well written with all the relevant data included in the manuscript or as supplemental data.*

## Experimental design

*The experimental design is very straightforward and satisfactory as IFN-g expression in response to specific peptide stimulation is a very common approach used towards understanding if a host was exposed to a specific pathogen.*

## Validity of the findings

*The findings are very relevant to the possible strategy for utilizing malaria vaccines as they become available for testing. Conclusions are reasonable, given the data and appropriate speculation about the possible need for boosters is included.*

## Comments for the Author

*This is a very straightforward paper that investigates the memory of a response to malaria antigens during a period of low malaria transmission. The data is straightforward and consistent with a model that would possibly require booster shots that would strengthen the host immune response. I only have a few comments:*

1. *The authors state the lack of any sex bias but did they compare the sexes within the different age groups?*

**It is true that our assertion is based on preliminary testing across all age groups, without comparing sexes within each age group. However, we have now repeated the tests within age group and found no bias within the ages (see table below showing p values obtained by comparing IFN- γ responses by gender, using t-test)**

| Comparing Levels of IFN-γ responses by gender in different Age- groups. Table is based on April 2008 round. | | | | |
|---|---|---|---|---|
| | **<5yrs age** | **6-14yrs** | **15-40** | **>40yrs** |
| **AMA1** | **P=0.48** | **P=0.53** | **P=0.23** | **P=085** |
| **CSP** | **P=0.19** | **P=0.58** | **P=0.68** | **P=0.31** |
| **LSA1** | **P=0.31** | **P=0.78** | **P=0.53** | **P=0.27** |
| **MB2** | **P=0.20** | **P=0.89** | **P=0.61** | **P=0.23** |
| **MSP1** | **P=0.19** | **P=0.92** | **P=0.51** | **P=0.08** |
| **TRAP** | **P=0.39** | **P=0.54** | **P=0.53** | **P=0.21** |

2**.** *The population was healthy at the start but were there any later infections that might have impacted the results?*

**This study involved weekly follow-up of study participants for any signs of malaria over the study period; and none reported any fever or signs of any chronic infection. Although we did not test for helminthic infections, there was no intervention over the study period that would have altered their levels in the population. The most likely explanation for the results was interruption of malaria**

**transmission.**

3. *How were cutoff IFN levels for individual peptides determined?*

**The cut-off IFN levels was determined by stimulating PBMC from 13 malaria naïve North Americas with each antigen then testing for IFN-γ levels in the culture supernatants by ELISA. For each antigen, the mean IFN-γ levels + 2 standard deviations was then determined. We have clarified this in the manuscript (Line 190-192).**

# Reviewer 2 (Anonymous)

## Basic reporting

*There are a number of typographical errors in the manuscript – it is suggested that the authors carefully re-read their manuscript and fix these errors to improve the language and readability. The introduction is otherwise well written, clear and easy to understand. Past literature has been adequately cited, although it is recommended that the more recent version of the WHO world malaria report be used. The structure of the manuscript confirms to PeerJ standards. The figures and tables are relevant; some suggestions for improvement have been made (see General Comments). The raw data has been provided, however further information should be included to make this easier to understand i.e. include another sheet in the excel files that provides a description for what each of the variables/headers means.*

**Agreed. We apologize for this omission. We have now revised the raw data file and included a sheet named "Log" with a description for each variable.**

## Experimental design

*Overall, the study has a clear research question on the longevity of antigen-specific IFN-y responses following interrupted malaria transmission. This topic will be of interest to the malaria immuno-epidemiology and vaccine fields.*

*It would have been preferable if the authors had tested IFN-y levels for each individual at all time-points on the one plate. They have tried to account for this by measuring the batch effect. However, they have failed to discuss the results of this experiment. Given its importance this should be discussed, even if just briefly in the results section.*

**Agreed. We have included a statement (Line 277-280) on the interpretation of the results**

*Half-lives have been calculated from a start-point when there had already been no malaria exposure for approximately one-year. Hence the IFN-y responses may have already declined significantly since the last exposure. This may result in some variability of the estimate compared to the true IFN-y half-life responses for these antigens.*

**Agreed. We have  introduced a statement in the discussion to clarify this as a weakness of the study.**

*I think the methodology for obtaining the estimated half-lives should be introduced in more detail, either in the materials and methods or results sections.*

**We have added a brief description of the methods for the half-life estimates and confidence intervals.  The referee did not indicate how much detail was desired;  we hope we have struck an adequate balance between informativeness and brevity.**

## Validity of the findings

The conclusion of the manuscript is well supported by the authors' findings, and the findings themselves are valid.

## Comments for the Author

*Introduction*

*Line 67-69. If studies have suggested the breadth and magnitude of IFN-y responses reduce quickly upon resolution of infection, it would suggest to me that IFN-y responses are more markers of current or recent exposure, not past exposure (as stated in Line 69). 'Past exposure' does not define very well how far in the past you mean.*

**Agreed. We have amended the section to read "..past-exposure"(line 71).**

*Materials and methods*

*Line 113. Should include the ethics approval numbers.*

**We have included KEMRI Approval number (line 115)**

*Line 121. Define PBMC, peripheral blood mononuclear cells.*

**We have defined PBMC (line 122)**

*Line 122. Also state the median/mean age.*

**We included both median/mean age (line 123)**

*Line 125. Was clinical malaria assessed weekly during the entire study period, i.e. April 2008-April 2009? This is not clear. Were the individuals with symptoms (referred to the local health facilities) tested for malaria, or only at the three visits?*

**Yes, clinical malaria was assessed weekly over the study period. However, none of the participants was found with malaria symptoms during weekly visits.**

*Line 138. Define PHA. I don't think the details on PBMC culturing are needed here, given this is explained in detail in section 2.5.*

**PHA has been defined and section on PBMC culture deleted.**

*Line 145. Define MB2. Which antigen is this?*

**MB2 protein is a Plasmodium antigen expressed on sporozoite surface antigen. (Nguyen et al, 2009 Malar J, Ochola et al, 2015 ).**

*Section 2.3. Would the peptide information be simpler to read in a table format?*

**Agreed. We transferred the information into Table 1**

*Results*
*Line 225. I assume this refers to 'prevalence of seropositivity' of IgG antibodies to CSP remained unchanged. This should be clarified.*

**Yes, the line referred to seropositivity. We have revised to make it clear (line 244-246).**

*Line 228. I could not locate supplementary table 1 in the materials provided.*

**We apologize for the omission. We have now included the Table.**

*Figure 2. Suggest adding in the positivity cut-offs as a line on each graph.*

**Agreed. We have added a cut-off line in each graph**

*Table 1. The estimated half-lives could be given in days or months, given they are all less than 1 year.*

**We have converted half-life values to days**
*Discussion*

*Line 276. The results of the antibody ELISA measurements should be mentioned here, given the reason for doing this was to support the assumption that there was no/limited exposure to malaria parasites during the study period.*

**Agreed. We have added a statement on the antibody Elisa (line 304-307)**

*Line 305. It is interesting that the responses decreased most dramatically in the first 6 months of the study. This potentially suggests that the decay in IFN-y responses is not simply linear, but maybe biphasic, as has been shown for antibody responses (i.e. White MT et al 2014 BMC Med).*

**Agreed. We have added a statement in the discussion (334-336).**

*Line 310. Were any of the same samples used? It would be interesting to compare the IFN-y responses tested using the two different assays in the same population.*

**Yes, some individuals were common in both tests. We will take the suggestion and analyze responses by the two assays in a subsequent paper.**

*Line 330. Review this sentence: 'It is notable…'. The grammar needs to be checked to make sure the meaning is clear.*

**Agreed. We have revised accordingly.**

*Line 335. Could this also be an effect of comparing individuals from non-malaria endemic areas and malaria endemic areas? For example, often vaccines perform better in trials of individuals from non-malaria endemic regions.*

**Agreed. Differences in malaria exposure histories may also account for the differences.**

# Reviewer 3 (Anonymous)

## Basic reporting

See comments below.

## Experimental design

See comments below.

## Validity of the findings

See comments below.

## Comments for the Author

Ayieko et al. aimed to test the specific IFN-gama response in a big cohort of individuals living in an area where malaria transmission was interrupted for more than a year. Three blood collections were performed in one year-interval, and the IFN-g response against six different Plasmodium antigens derived from pre-erythrocytic and erythrocytic phases was tested. During the study period, the incidence of malaria in the area was extremely low (less than 1%). The authors show a significant drop in the % of specific IFN-g positive responses over time.

The paper is well written and is easy to follow. The results add new information on specific IFN-g immune responses in areas where Plasmodium falciparum malaria transmission has been interrupted. This is relevant because IFN-g production has been correlated with protection against malaria, and a drop on this response may lead to increased risk of malaria acquisition, especially in the context of an anti-malaria vaccine. It is important to mention that the authors carefully discuss some articles that reached different conclusions and try to point out the differences among the studies. Overall, this paper adds interesting information on the dynamics of IFN-g responses in areas that may have malaria transmission interrupted for some time.

Minor comments:

1. I could not find Supplementary Table 1 in the submitted material.

**We have now provided the information**.

*2. ELISAs were performed with CSP and schizont extract. Please provide information on both. Which CSP was used? Recombinant protein produced in bacteria? From which Plasmodium clone? What was the concentration added per well? And how was the schizont extract prepared? What was the concentration used in the ELISAs? This information should be added to the materials and methods section.*

**We have provided the information on CSP and schizont extract.**

## *Technical changes*

*These are your technical changes from PeerJ staff:*

*# Acknowledgments*
*Please remove permissions information from the Acknowledgments and include it in your Methods section.*
**Agreed. Removed as suggested**.

*# Data not Shown*
*We noted your statement "Data not Shown" (in line 265) and we would like to draw your attention to our Data Sharing policy as detailed at <https://peerj.com/about/policies-and-procedures/#data-materials-sharing>. Of course, the inclusion of this statement does not necessarily mean that our policy is being violated, so please can I ask you to leave a note to staff at <https://peerj.com/manuscripts/12944/declarations/#other> or email me (at editorial.support@peerj.com) to let me know the reason(s) for not showing this data in this instance?*

**We have now revised Figure 3 to include all the data for October 2008 and April 2009, which was previously not shown. We did not previously show the data because we the trend was generally similar to April 2008.**

*# Consent Form*
*Please provide an empty copy of the human participant consent form you used as a*

*confidential Supplemental File here <https://peerj.com/manuscripts/12944/files/>.*

**We have attached the form**

*# Competing Interests*

*Please remove all competing interests information from the source file manuscript and make sure it is included in your Competing Interest Statement instead here <https://peerj.com/manuscripts/12944/declarations/#question_17>.*

**We have removed the information**

*# Affiliations*

*1) We notice that the author affiliations you have provided in the system are slightly different to those in the document.*

**We have revised the section accordingly**

*2) As our system will treat these affiliations as metadata, please ensure that both the 'system version' and the 'document version' are complete and the same.*

**We have revised this accordingly.**

*3) Please edit the author affiliations using the 'Edit' button to the right of the names here <https://peerj.com/manuscripts/12944/authors>, or edit your manuscript source file and upload it here <https://peerj.com/manuscripts/12944/files>.*

*# Manuscript Source File*

*1) Please provide the clean unmarked source file (e.g. .DOCX, .DOC, .ODT) with no tracked changes shown, all tracked changes accepted and tracked changes turned off.*
***DONE***

*2) Please upload the manuscript file in the Revised Manuscript & Primary Files section here: https://peerj.com/manuscripts/12944/files/.*

*3) If you uploaded a PDF because of formatting problems, please provide the source file as a Supplemental File and we will mark it as the correct file type as necessary if the manuscript is accepted.*